# Does Siponimod Exert Direct Effects in the Central Nervous System?

**DOI:** 10.3390/cells9081771

**Published:** 2020-07-24

**Authors:** Markus Kipp

**Affiliations:** Institute of Anatomy, Rostock University Medical Center, Gertrudenstrasse 9, 18057 Rostock, Germany; markus.kipp@med.uni-rostock.de; Tel.: +49-381-494-8401

**Keywords:** sphingosine-1 phosphate, multiple sclerosis, inflammation, siponimod, astrocytes, demyelination, neuroprotection, astrocytes, microglia, fingolimod

## Abstract

The modulation of the sphingosine 1-phosphate receptor is an approved treatment for relapsing multiple sclerosis because of its anti-inflammatory effect of retaining lymphocytes in lymph nodes. Different sphingosine 1-phosphate receptor subtypes are expressed in the brain and spinal cord, and their pharmacological effects may improve disease development and neuropathology. Siponimod (BAF312) is a novel sphingosine 1-phosphate receptor modulator that has recently been approved for the treatment of active secondary progressive multiple sclerosis (MS). In this review article, we summarize recent evidence suggesting that the active role of siponimod in patients with progressive MS may be due to direct interaction with central nervous system cells. Additionally, we tried to summarize our current understanding of the function of siponimod and discuss the effects observed in the case of MS.

## 1. Introduction

Multiple sclerosis (MS) is a chronic inflammatory disease that affects the central nervous system (CNS), especially the brain, spinal cord, and optic nerve. Although some people have mild symptoms, such as blurred vision, numbness, or tingling of the limbs, in severe cases, the patient may experience paralysis, vision loss, or cognitive deficits. Although we do not know exactly what causes MS, T- and B-cell-mediated inflammatory demyelination of the CNS white and gray matter, parallel to neuro-axonal damage, are histopathological features. Clinically, MS can be divided into four major categories: clinically isolated syndrome (CIS), which refers to the first episode of neurological symptoms that lasts at least 24 h, but the criteria for diagnosing MS have not (yet) been met. Individuals who have experienced CIS may or may not continue to develop definitive MS. Relapsing-remitting MS (RRMS) is characterized by an acute clinical attack, followed by complete or incomplete recovery, and a certain remission period between attacks. A few years later, many patients with relapsing-remitting disease eventually develop secondary progressive MS (SPMS), which is characterized by a more or less continuous decline in neurological function, with or without occasional attacks. Primary progressive MS (PPMS) is characterized by the accumulation of clinical disability from the onset of the disease, without early relapses or remission. In the stage of RRMS disease, the most important indicator of successful therapeutic intervention is to reduce the frequency of relapses. On the contrary, during the progressive disease stage, treatment aims to delay the progression of the disability. Although there are multiple options for treating RRMS patients, only two drugs, siponimod [1] and ocrelizumab [2], have shown therapeutic effects and have been approved for the treatment of progressive MS. It is worth noting that most phase II and III clinical trials of progressive MS have not shown any beneficial effects, which is the main requirement for a better understanding of the underlying progressive MS pathology.

As will be pointed out later, siponimod (Mayzent^®^ and Novartis) is a sphingosine 1-phosphate receptor (S1Pr) modulator that can selectively bind to S1Pr1 and S1Pr5. These receptors are expressed by various neuronal and peripheral cell populations, such as lymphocytes, dendritic cells [3,4], astrocytes [5,6], microglia [7,8], and oligodendrocytes [5,9]. In this review article, we try to summarize our current understanding of the cellular functions of siponimod and discuss the effects observed in the context of MS. Firstly, we focus on the role of glial cells in the development and progression of MS disease, then briefly introduce the sphingosine 1-phosphate signaling cascade, and, thirdly, discuss the step how siponimod ultimately interferes with these cell populations to exert beneficial effects.

## 2. General Aspects of MS Pathology and the Relevance of Glial Cells

It is generally accepted that the histopathological correlate of an initial attack or relapse is the focal inflammatory demyelinating lesion. This focal lesion is mediated by an aberrant lymphocyte attack against CNS elements. Although the autoantigens are still unknown, it is believed that the autoreactive immune response of MS patients is directed against a particular component (or multiple components) of the myelin sheath. These inflammatory lesions can be widely found in white matter areas [10], and the neuroanatomical location of the lesion can cause specific clinical symptoms. For example, optic nerve tract lesions may cause vision loss and cerebellar lesions may cause coordination problems or ataxia of the limbs, gait, and trunk, while lesions in the corticospinal tract can be related to motor dysfunction [11]. Under the microscope, focal lesions show peripheral immune cell recruitments (mainly macrophages and CD8^+^-lymphocytes), a loss of myelin and oligodendrocytes, the intense activation of astrocytes and microglia, and acute axonal injury (see Figure 1). At the paraclinical level, oligoclonal immunoglobulin G in cerebrospinal fluid and gadolinium-enhanced lesions on magnetic resonance scans are considered to be the direct result of focal inflammatory CNS lesions. Although not the focus of this article, several studies have shown that, during RRMS, not only white matter regions are affected but, also, various structures of the brain gray matter [12,13].

Contrary to the focal features of RRMS, the pathologies of SPMS and PPMS are more diffuse. For example, the brains of progressive MS donors show a significant axonal loss in both the demyelinated and normal cortices [14], extensive subpial grey matter demyelination [10], injury to excitatory projection neurons [15], diffuse neuroaxonal metabolic abnormalities [16] and diffuse microglial activation [17]. The underlying mechanisms of these diffuse CNS pathologies in progressive MS are complex but may include transneuronal degeneration due to the destruction of efferent/afferent projections [18], B cell-rich meningeal inflammation [19], the accumulation of peripheral immune cells in the choroid plexus stroma [20], chronically active and slowly expanding lesions with smoldering inflammation [21], accelerated biological aging [22], or complement activation [23]. Different aspects of the described progressive MS pathologies can be visualized by different imaging techniques, including positron emission tomography (PET) imaging, advanced magnetic resonance imaging (MRI), brain volume measurement, optical coherence tomography (OCT), diffusion tensor imaging (DTI), or myelin water imaging (MWI) [24,25,26,27,28]. 

In the course of MS disease, myelin-axonal units are progressively destroyed. As shown in Figure 2A, demyelination is accompanied by acute axonal defects, such as the breakdown of the anterograde axonal transport mechanism and the accumulation of intra-axonal organelles at sites of inflammation. In addition to peripheral immune cells (such as lymphocytes and macrophages), astrocytes and microglia are also important regulators of myelin and neuronal injury in MS. Astrocytes (literally “star-like cells”) are the largest number of glial cells in the CNS. They play an important role in development, health, and disease. For example, they maintain brain homeostasis through the expression of neurotransmitter transporters; store and distribute energy substrates; control the development of neural cells, synaptogenesis, and synaptic maintenance; and provide cytokines and limiting membranes for brain defense. During CNS injuries, astrocytes are activated, which can be identified by the strong upregulation of the glial fibrillary acidic protein (GFAP; see Figure 2C), and a series of changes occur, called reactive astrogliosis. Additionally, during inflammatory stimulation, astrocytes induce the expression of hundreds of genes relevant to antigen presentation, oxidative stress, immune receptors, inflammation, blood-brain barrier disruption, and signal transduction [29]. In the context of MS, astrocytes can provide promyelinating neurotrophic factors, such as ciliary neurotrophic factor [30], or tissue inhibitors of metalloproteinase-1 [31], orchestrate oligodendrocyte differentiation (recently reviewed in [32]), or stabilize the blood-brain barrier and, thus, limit the recruitment of peripheral immune cells [33] and CNS pathology [34]. Although these effects indicate that astrocytes have proregenerative or protective functions, deleterious astrocytic effects have also been shown, such as astrocyte-induced neurodegeneration [35], inflammation [36,37] and the recruitment of peripheral immune cells. Indeed, astrocytes display a wide range of selective responses, and multiple functional states that can promote lesion formation (i.e., proinflammatory phenotypes) or resolution (i.e., anti-inflammatory phenotypes) exist. For example, the laboratory of Francisco Quintana recently identified astrocytes in experimental autoimmune encephalomyelitis (EAE) and MS, characterized by the reduced expression of nuclear factor erythroid 2-related factor 2 (NRF2) and the high expression of MAF bZIP transcription factor G (MAFG), resulting in the repression of antioxidant and anti-inflammatory transcriptional programs in astrocytes [36]. Therefore, the downregulation of NRF2 expression appears to be a cellular event that promotes the shift from lesions resolving to lesion-promoting astrocytes. Consistent with this, the metabolic injury to oligodendrocytes that were experimentally induced via intoxication with cuprizone [38] is increased in *Nrf2^−/−^* compared to wild-type mice [39], while the activation of NRF2, especially in GFAP^+^ astrocytes, improved the pathology in a toxin-induced demyelination model [40]. 

Astrocytes are also key regulators of the brain-blood interface. Peripheral immune cells can invade the brain via three main neuroanatomical routes: firstly, at the level of the postcapillary venule (i.e., blood-brain barrier; BBB), secondly, from the choroid plexus stroma into the ventricle, and from there, into the CNS parenchyma (i.e., the blood-liquor barrier), and thirdly, along with the spaces between penetrating brain arteries (i.e., the Virchow-Robin space). Here, we will focus on the BBB, which acts as a super-selective filter for molecules and cells to protect the brain from the blood milieu [42]. The two main barrier functions of the BBB are (i) endothelial cells, which express unique intercellular tight junctions to seal the paracellular space, and (ii) highly specialized basement membranes, which are formed from endothelial cells and astrocytes (see Figure 2E). For the sake of completeness, it should be mentioned that the BBB also contains pericytes and perivascular macrophages. The results of several studies clearly showed that a close endothelial-astroglial association is necessary for the induction, organization, and maintenance of the BBB [43]. For example, removal of the astrocytes from in vitro BBB coculture models leads to increased paracellular permeability for small tracers across the brain endothelial cell monolayer [44]. Beyond, the expression of vascular cell adhesion protein 1 (VCAM-1) by astrocytes is crucial for the entry of T cells into the CNS parenchyma [45], the loss of astrocyte polarity is a characteristic of the impaired BBB [46], and the transgenic inactivation of astroglial NF-kappa B reduces the recruitment of peripheral immune cells in EAE [47]. Although we currently do not know whether neurotoxic astrocyte polarization can be equated with an astrocyte that facilitates the recruitment of peripheral immune cells, the modulation of astrocyte function by, for example, siponimod might be a promising therapeutic strategy. Remarkably, astrocytes as well express specific receptors to communicate with various lymphocyte subpopulations, such as Th1 and Th17 cells, as recently demonstrated [48]. Indeed, it is believed that astrocytes are the most subtle regulators of immunocompetent T cells and are central to the physiological immune reactivity of the CNS. It will be tempting to find out to what extent glial cells are involved in the transmission of regulatory signals between the immune system and the nervous system.

Microglia, which originate from myeloid precursors in the embryonic yolk sac, are like astrocyte highly plastic immune cells. During the RRMS disease stage, microglia are believed to be critical for myelin phagocytosis, T cell antigen presentation, and the release of proinflammatory cytokines. However, when the MS moves into the progressive phase, it is believed that microglia play an important role in the slow expansion of chronic lesions, one of the pathological features of progressive MS (see Figure 1). Microglia exist, in the same way as astrocytes, in a continuum of activation states and can, therefore, be involved in both tissue injuries and repairs. On the one hand, it has been shown that microglia mediate synapse loss, neuronal loss, and memory impairment in Alzheimer’s disease (overview in [49] and [50]); mediate dopaminergic injuries via the NF-κB signaling pathway in Parkinson’s disease [51]; or provide an environment for the initiation of T-cell cytotoxicity in Rasmussen encephalitis [52]. On the other hand, microglia can equally mediate protective effects such as reducing spreading depolarization and calcium overload [53], eliminating neutrophil invasion in brain ischemia [54], promoting seizure-induced neurogenesis [55], or maintaining the vascular integrity under hypoxia conditions [56]. In connection with MS, it has been shown that phagocytosis of myelin debris by the microglia is a prerequisite for endogenous remyelination [57,58], but the cells can also induce a core oxidative stress gene signature when activated, which leads to axonal damage [59]. Remarkably, astrocytes and microglia interact closely, as several studies have shown [60]. For example, a recent study showed that the drug-induced prevention of microglia-mediated astrocyte conversions to a neurotoxic phenotype is protective in a model of Parkinson’s disease [61]. The results from Martin Stangel’s laboratory suggest that astrocytes can regulate the clearance of myelin debris by activating and/or attracting microglia in a CXCL10-dependent manner [62]. Interestingly, aging is associated with remyelination failure and is accompanied by a decrease in the ability of microglial cells to phagocytose myelin. As was recently shown, the reduced expression of the scavenger receptor CD36 could be the reason for this reduced phagocytic activity [63]. In summary, astrocytes and microglia cells are important regulators of the pathology of MS diseases, and it is promising to shape their function through therapeutic intervention. 

## 3. Sphingosine-1 Phosphate Signaling

Sphingosine-1-phosphate (S1P) is a bioactive sphingolipid that regulates a wide range of physiological processes, including lymphocyte recirculation, cardiac function, or the maintenance of the BBB [64]. Most S1P effects are mediated through one of the five G protein-coupled S1P receptor subtypes called S1Pr1–5 (originally called EDG-1, 3, 5, 6, and 8) [65]. These receptors are expressed differently on different cell types, including lymphocytes [66,67], cardiomyocytes [68,69], endothelial cells, smooth vascular muscle cells, or fibroblasts. 

The coordinated migration of cells is important for various biological processes, including wound healing, embryonic development, and immune responses. Initially, S1Pr signal research focused on the migration of progenitor cells. In 2000, Kuppermann and colleagues published a sentinel paper that showed that S1Pr is crucial for cell migration during cardiovascular development [70]. In zebrafish, as with all vertebrates, the embryonic heart is created as a bilateral cell group in the anterior lateral plate mesoderm. These two cell populations move medially and merge along the midline to form the primitive heart tube [71]. In their studies, they first identified eight mutations that disrupt heart development, including a mutation called miles apart m (93), also known as Mil or Milm 93, through large-scale genetic tests in zebrafish. Using zebrafish carrying the miles apart m (93) mutation, they observed normal myocardial differentiation but disrupted myocardial precursor migration, indicating that separate signaling pathways regulate myocardial differentiation and migration. Using genetic mosaics, mutant cells transplanted into wild-type embryos migrated normally and contributed to the heart, while wild-type cells transplanted into mutant embryos did not migrate properly. This elegant experiment has shown well that cells other than myocardial progenitors must express the wild-type protein to orchestrate the migration of myocardial progenitor cells (i.e., a non-cell-autonomous process is operant). Based on these findings, it has been speculated that wild-type Mil could stimulate the release of a chemotactic factor or create an environment that supports progenitor cell migration. In subsequent experiments, the authors were able to show that Mil is a G protein-coupled receptor, that the abovementioned miles apart m (93) mutation means that Mil is unable to attach to downstream G proteins, and that Mil can respond to S1P. Today, we know that Mil is most likely the S1Pr2 in humans and mice [72]. 

The S1Pr signaling not only regulates cell migration during the heart but, also, vascular development. During early development, S1Pr1^−/−^ mice show gross defects in their arterial and capillary vessels. Remarkably, both vasculogenesis and angiogenesis begin normally in mutated S1Pr1 embryos, which leads to well-developed endothelial cell networks. The faulty process in S1Pr1^−/−^ embryos is the migration of cells that lead to support structures that surround arterial and capillary blood vessels, the smooth vascular muscle cells, and pericytes [73]. The regulation of cell migration by S1Pr is not limited to the development phase but regulates important physiological processes in adulthood. From an immunological point of view, Norgauer’s laboratory showed that S1P induces the chemotaxis of immature dendritic cells and promotes the release of Th2-related cytokines in mature dendritic cells, which favors immunity dominated by Th2 lymphocytes [3]. At the same time, Graeler et al. showed that CD4-Th and CD8 cytotoxic lymphocytes express S1P receptors (predominantly S1Pr1 and S1Pr4) and that S1P causes chemotactic reactions in T cells [66]. Finally, Mandala et al. showed that the functional antagonism at the S1P receptor blocks the escape of lymphocytes from the lymph nodes [74] in a S1Pr1-dependent manner [75]. It was subsequently shown in various experimental environments that this antimigratory effect leads to therapeutically useful immunosuppression, including in models of antigen challenges [76], graft-versus-host disease [77], respiratory tract infection [78], transplantation experiments [79,80,81], experimental colitis [82], type 1 diabetes [83], experimental arthritis [84], systemic lupus erythematosus [85], or, especially, in the context of MS research in various EAE models [86,87], the autoimmune model of MS. Today, we know that the migration of even neuronal cells such as astrocytes can be regulated by S1Pr signaling [88]. 

Most of these studies used FTY720 (2-amino-(2-[4-octylphenyl]ethyl)-1,3-propanediol hydrochloride; also called fingolimod) to modulate S1Pr activity, and FTY720 was the first effective S1Pr-modulating drug to be used in RRMS patients [89]. Fingolimod (Gilenya^®^ and Novartis) was approved in 2010 as the first oral treatment for RRMS in several countries. In 2018, Novartis announced that the U.S. Food and Drug Administration (FDA) approved Gilenya^®^ for the treatment of children and adolescents aged 10 to under 18 with RRMS. It is the first disease-modifying therapy that is indicated for these patients. Although originally described as an S1Pr agonist, it has now become generally accepted that fingolimod acts as a functional S1P inhibitor by inducing S1Pr internalization and intracellular partial degradation [90]. It should be noted that S1P signaling is not only important for T- but, also, B-cell functions. For example, it was shown that S1Pr1 is required for the correct positioning of B cells within the spleen [91,92]. The presence of (B cell-rich) lymphoid follicle-like structures in the meninges of some MS patients [93,94] and the clinical efficacy of the B-cell depleting antibody ocrelizumab [2] suggest that B cells contribute to the course of the disease in MS. To what extent the S1Pr signal transmission also coordinates the formation or dissolution of B-cell follicle-like structures in MS is yet to be clarified. 

The natural ligand of S1Pr, sphingosine-1-phosphate, is derived by phosphorylation of the membrane lipid sphingosine, a reaction catalyzed by type 1 or 2 sphingosine kinase (Sphk1/Sphk2). Sphk2 is the dominant isoform that catalyzes S1P synthesis in the CNS [95], protects against ischemic brain damage in stroke models [96], or maintains the plasticity of the hippocampus through the S1P-mediated inhibition of histone deacetylases [97]. Just like sphingosine-1-phosphate, fingolimod/FTY720 has to be activated by phosphorylation to the biologically active metabolite FTY720-P, which is mainly catalyzed by Sphk2 [98,99]. Beyond, the activities of lipid phosphate phosphohydrolase 3 (LPP3), also known as phospholipid phosphatase 3 (PLPP3) or sphingosine-1-phosphate phosphohydrolase (SPP1), can convert FTY720-P into its inactive, dephosphorylated metabolites [100] and, thus, reverse the mechanism actions from Sphk2. In several studies, the dynamic regulation of sphingosine kinases and phosphatases was observed in some disease models such as Alzheimer’s [101] or brain tumors [102]. To what extent the expression of these enzymes changes in MS and how this may affect the therapeutic efficacy of fingolimod remains to be determined. Although this is not the key enzyme for FTY720 phosphorylation, the upregulation of Sphk1 expression in astrocytes and macrophages/microglia has been found in MS lesions [103]. Beyond, it has been shown that S1Pr expression levels dynamically change during the formation of inflammatory lesion in MS, such as increased S1Pr1 and S1Pr3 expression levels on reactive astrocytes in active and chronic inactive MS lesions [104], indicating that astrocytes may act as target of fingolimod and siponimod within the CNS. Induced S1Pr expression levels have as well been reported in the EAE model [105,106,107]. Interestingly, it has been suggested that overexpression of the S1Pr1 on reactive astrocytes drives the neuropathology of the MS rebound after fingolimod discontinuation [108]. 

## 4. From FTY720 to Siponimod

In 2013, Selmaj et al. reported the results of a phase 2 dose-finding study in patients with RRMS. Siponimod reduced the number of active brain lesions and the annualized relapse rate by around two-thirds, depending on the dose [109]. Based on these promising results, a placebo-controlled phase III trial was conducted, in which the efficacy and safety of siponimod in patients with SPMS were examined (Examination of the efficacy and safety of siponimod in patients with secondary progressive multiple sclerosis (EXPAND)) [1]. The study showed that siponimod is clinically effective in SPMS patients based on the primary outcome of a three-month confirmed disability progression reduction (defined by a 0.5 or 1 point increase in the expanded disability status scale (EDSS) compared to the baseline). The key secondary objective was achieved with a 26% reduction in the confirmed disability progression after six months, and a significant reduction in the annualized relapse rate and MRI activity were noted. Using a matched-adjusted indirect comparison, it was shown that siponimod is significantly more effective compared to interferon-beta therapies for the result of up to six months of confirmed disability progression (CDP) [110]. In March 2019, siponimod was approved in the United States for the treatment of adults with MS, including patients with CIS, RRMS, and active SPMS. In January 2020, siponimod was approved in the European Union for the treatment of adults with SPMS with an active disease that has been demonstrated by relapses or imaging features of inflammatory activity. It is noteworthy that fingolimod, the first-generation S1Pr modulator, was ineffective in the phase III INFORMS trial for primary progressive MS (i.e., Oral fingolimod in primary progressive multiple sclerosis) [111]. 

There are several important differences between FTY720 and siponimod. Firstly, the chemical structure of both drugs is different (see Figure 3). While fingolimod is an aminodiol consisting of propane-1,3-diol with amino and 2-(4-octylphenyl)ethyl substituents at the 2-position, the chemical structure of siponimod is much more complex [112]. Secondly, while fingolimod is a prodrug that needs to be activated by Sphk2 (see above), siponimod does not require activation. Thirdly, while fingolimod binds to four of the five S1P receptors—namely, S1Pr1, S1Pr3, S1Pr4, and S1Pr5—siponimod predominantly interferes with the two receptor isoforms S1Pr1 and S1Pr5. 

Clinically, the first dose of fingolimod is associated with a decrease in heart rate and a slowdown in atrioventricular conduction [113,114,115]. In 2012 and 2013, Pan and Gergely et al., from the Genomics Institute of the Novartis Research Foundation (San Diego, CA, USA) and the Novartis Institute for Biomedical Research (Basel, Switzerland), described the rationale and the procedure of siponimod development [112]. The basic principle was to develop a compound that spared the S1Pr3 receptor subtype, since it was believed to be responsible for the bradycardia observed based on the lack of S1P-induced heart rate reduction in S1Pr3 knockout mice. Siponimod has also been developed to have a relatively short elimination half-life, which enables the rapid restoration of the blood lymphocyte count after treatment has ended but enables once-daily oral dosing [116]. At the cellular level, the authors observed that (i) siponimod is a selective modulator on S1Pr1 and S1Pr5 receptors and induces the profound and prolonged internalization of S1Pr1 receptors, (ii) that siponimod improves EAE in a therapeutic setting (i.e., suppresses ongoing disease symptoms), and (iii) that siponimod induces a dose-dependent reduction in the peripheral absolute number of lymphocytes in humans [116]. Of note, it has been shown that, in contrast to S1Pr1, S1Pr5 is not downmodulated by agonists such as siponimod or fingolimod. Consequently, S1P5 agonist function, and not functional antagonism, should be considered when studying the effects of siponimod and fingolimod mediated via S1Pr5. Contrary to expectations, the treatment with siponimod caused G protein-coupled inwardly rectifying potassium (GIRK) channel activation in human atrial myocytes and bradycardia in healthy volunteers, suggesting that this side effect is not mediated via S1Pr3 in humans (in contrast to mice). In this context, it is important to notice that, at least for fingolimod, S1Pr-indeendant effects have been demonstrated [117]. 

The observed anti-inflammatory potency of siponimod in EAE has since been reproduced by others [118]. In a recent study, the authors showed that the adoptive transfer of proteolipid protein–primed Th17 cells into SJL/J recipient mice induces subpial demyelination, microgliosis, and destruction of the glial limitans superficialis and that this inflammatory cortical demyelination is improved by siponimod treatment [119]. Mechanistic experiments showed that S1Pr1/5 signaling is required to optimize the formation of meningeal tertiary lymphoid tissue in the subarachnoid space because of siponimod-attenuated fibronectin formation. In SPMS patients, RNA derived from whole-blood samples of siponimod-treated patients have reduced expression levels of immune-associated genes involved in T- and B-cell activation and receptor signaling, which is consistent with the reduction in CD4+ T cells, CD8+ T cells, and B cells. Flow cytometric analyses showed that, within the remaining lymphocyte subsets, the incidences of CD4^+^ and CD8^+^ naive T cells were reduced, while anti-inflammatory Th2 and T regulatory cells (Tregs) were enriched [120], indicating a shift towards an anti-inflammatory and suppressive homeostatic immune system that can contribute to the clinical effectiveness of siponimod in SPMS. 

As already mentioned, S1Pr1 and S1Pr5 are also expressed by cells of the CNS, including astrocytes [104], oligodendrocytes [121,122], microglia, or neurons [7,123]. It is therefore of great interest to know whether the drug can have beneficial effects in MS that are not mediated by immunosuppression. To address this issue, siponimod was delivered directly to the brain using a continuous intracerebroventricular infusion. While this route of siponimod delivery improved the severity of EAE disease, the number of peripheral CD3^+^ cells was not affected [124]. Notably, astrocytosis, microgliosis, and neuronal degeneration were less severe in mice treated with siponimod, and interleukin 6 secretions were ameliorated in cultured microglia treated with siponimod. Additionally, using an organotypic slice culture model, it was shown that siponimod attenuates lysophosphatidylcholine-induced demyelination. At a more functional level, it has been shown that siponimod can improve cortical network functionality in acute brain slices isolated from EAE mice [118]. Since peripheral immune cells (especially lymphocytes) do not play a role in organotypic slice culture models, this experimental setup has shown convincingly that siponimod can have beneficial effects in the context of MS by directly modulating the brain cell function. Accordingly, the protective effect of siponimod is not only limited to EAE but has also been observed in other models such as intracerebral hemorrhage [125], where it limits the formation of perihemorrhagic edema, and in experimental stroke [126], where it limits peripheral immune cell recruitment, as well as in a mouse model of diffuse large B-cell lymphoma [127].

Direct interactions between siponimod and brain cells have also been demonstrated. For example, Gentile et al. observed that siponimod reduces the release of interleukin 6 and the chemokine CCL5/RANTES from activated microglial cells. Blocking the CCL5 or interleukin 6 receptor shows beneficial effects in EAE [128,129]. In a recent study, Colombo et al. demonstrated that induced pluripotent astrocytes from stem cells express the S1P receptor, show NF-κB translocation in response to exposure to interleukin 1 or interleukin 17, and that siponimod ameliorates NF-κB translocation. While glial cells exposed to these cytokines downregulated glutamate transporter protein expression, siponimod-treated astrocytes maintained high levels of the glutamate transporter. Remarkably, similar effects were observed when the cells were treated with FTY720. In the same study, the authors demonstrated that siponimod and FTY720 induced NRF2 nuclear translocation, suggesting that this important cellular antioxidant pathway could also be regulated by siponimod. Finally, coculture assays with induced pluripotent astrocytes from stem cells and spinal neuronal cultures showed that cytokine-stimulated astrocytes induced neurodegeneration, while this deleterious effect was improved by siponimod and FTY720. These results suggest that astrocyte targeting by S1P receptor modulators can save neurons from astrocyte-induced degeneration (i.e., shifting from lesion-promoting to lesion-resolving astrocytes).

Finally, it should be noted that S1Pr5 is also expressed by oligodendrocytes and their progenitor cells [121,130,131] and may, therefore, modify myelin repair (i.e., remyelination) but may also promote oligodendrocyte survival in an inflammatory environment. While the relevance of fingolimod for remyelination has been discussed in detail elsewhere [132], we consider it important to note that siponimod [133] was one of the most efficient substances in a Xenopus in vivo model among a range of molecules tested that favored remyelination. However, future studies with more complex organisms will be needed to test the potential remyelination function of siponimod.

## 5. Concluding Remarks

The potential advantages of using siponimod compared with fingolimod are its greater receptor specificity, more stable and predictable kinetics (due to the fact that siponimod is not a prodrug), and, most importantly, an approved beneficial effect of siponimod in a SPMS clinical trial. We believe that the current in vivo and in vitro evidence suggests that the effectiveness of siponimod in MS is due to additional direct effects in the CNS. The orchestration of oligodendrocyte maturation, together with indirect effects mediated by the function of astrocytes and/or microglia, appears to be operant. The protective effects of fingolimod and siponimod in MS and its preclinical models show that the sphingosine-1-phosphate pathway is an attractive target in both RRMS and progressive MS. The ultimate goal would be to understand the relevance of each receptor subtype to specific histopathological MS entities such as oligodendrocyte degeneration, demyelination, axonal injury, or BBB stability and, in a second step, to develop specific modulators that act on these receptors. In the future, new imaging techniques such as positron emission tomography (PET), single-photon emission computed tomography (SPECT), bioluminescence imaging (BLI), and fluorescent molecular tomography (FMT) could be used to visualize such histopathological entities in patients. 

## Figures and Tables

**Figure 1 cells-09-01771-f001:**
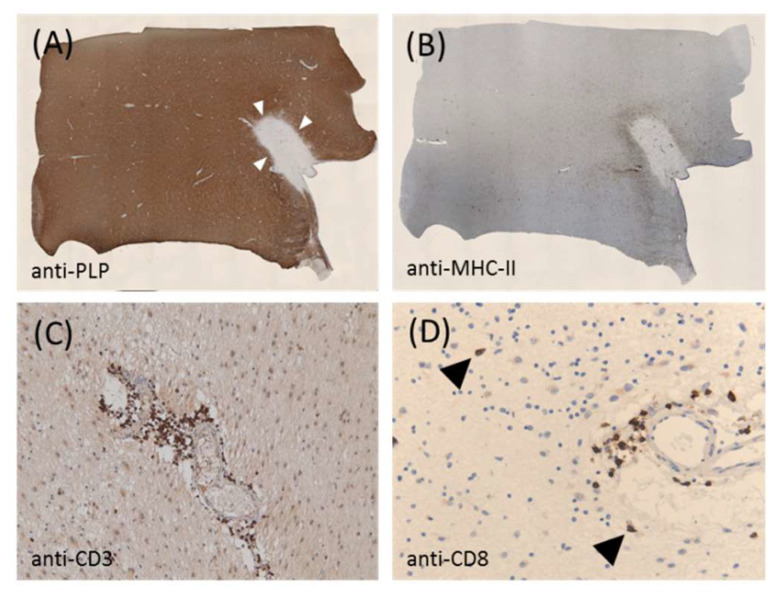
Shows a representative chronic active lesion from a patient with progressive multiple sclerosis (MS). (**A**) Shows anti-proteolipid protein-stained sections obtained from a white matter tissue block of a patient with progressive MS. The white arrowheads mark the boundary of the lesion. (**B**) Shows the same lesion processed for anti-MHC II immunohistochemistry. (**C**,**D**) Show the anti-CD3 and anti-CD8 staining of the lesion at higher magnification, respectively. The arrowheads highlight the cytotoxic CD8^+^ lymphocytes in the brain parenchyma. The paraffin-embedded postmortem brain tissue was obtained through a rapid autopsy protocol from donors with mainly progressive MS in collaboration with the Netherlands Brain Brank, Amsterdam, The Netherlands. The study was approved by the institutional ethics review board, and all donors or their relatives provided written consent for the use of brain tissues and clinical information for research purposes.

**Figure 2 cells-09-01771-f002:**
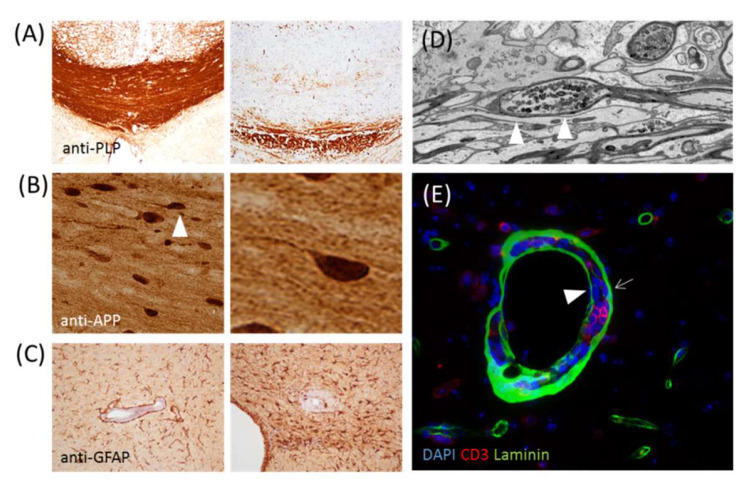
(**A**) Shows anti-proteolipid protein staining of the control corpus callosum (left) and corpus callosum of a cuprizone-intoxicated (right) mouse. (**B**) Shows a section stained with the anti-amyloid precursor protein of a mouse intoxicated with cuprizone. The axonal spheroid highlighted by the arrowhead is displayed on the right site at a higher magnification. (**C**) Shows the anti-glial fibrillary acidic protein-stained sections of a cuprizone-experimental autoimmune encephalomyelitis (EAE) mouse [41]. The image on the left shows moderate, and the image on the right shows severe, astrogliosis. (**D**) Shows the ultrastructure of an axonal spheroid. (**E**) Shows a perivascular inflammatory infiltrate stained with anti-CD3 and anti-laminin to label T-lymphocytes and basement membranes, respectively. The arrow highlights the astrocyte basement membrane, while the arrowhead highlights the endothelial basement membrane.

**Figure 3 cells-09-01771-f003:**
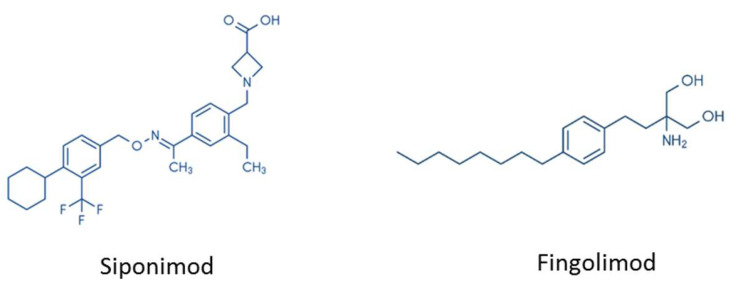
Chemical structures of siponimod and fingolimod.

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
