# Peer review of "Does Siponimod Exert Direct Effects in the Central Nervous System?"

_cells, 2020, doi:10.3390/cells9081771_

Round 1
Reviewer 1 Report
In this review article, the author described the role of S1P receptor modulator siponimod in the treatment of multiple sclerosis and speculated that its signaling in various CNS cells might contribute to the therapeutic effects. A number of issues need to be addressed:
- Since siponimod can act as a S1P receptor agonist or antagonist, what is the major downstream signaling leading to the immune suppressive and neuroprotective effects? Does the therapeutic dose of siponimod cause internalization of S1P receptors?
- Is there an upregulation or a differential expression of S1P 1-5 receptors in the EAE/MS tissues?
- Does the tolerance develop after prolonged siponimod treatment?
- Is S1Pr1 the dominant receptor attributing to the therapeutic effects of siponimod? More supporting evidence should be provided.
- What is the advantage of using siponimod compared with FTY720?
- What are the common adverse effects of siponimod in MS patients?
- Please use tables to summarize the findings of siponimod action in various cell types, as well as its therapeutic effects in the MS animal models and human patients.
- Which brain area is shown in Figure 1? Is there any reference (s) for the images presented?
Author Response
Q1:Since siponimod can act as a S1P receptor agonist or antagonist, what is the major downstream signaling leading to the immune suppressive and neuroprotective effects? Does the therapeutic dose of siponimod cause internalization of S1P receptors?
A1: Thank you for this excellent comment. We now state the following “Of note, it has been shown that in contrast to the S1Pr1, the S1Pr5 is not down-modulated by agonists such as siponimod or fingolimod. Consequently, S1P5 agonist function, and not functional antagonism, should be considered when studying the direct neuroprotective effects of siponimod.”Beyond we state that the NFκB and NRF2 signaling cascade have been identified as two major downstream signaling leading to the observed neuroprotective effects.
Q2: Is there an upregulation or a differential expression of S1P 1-5 receptors in the EAE/MS tissues?
A2: Thank you for this excellent comment. We now state “Beyond, it has been shown that S1Pr expression levels dynamically change during the formation of inflammatory lesion in MS, such as increased S1Pr1 and S1Pr3 expression levels on reactive astrocytes in active and chronic inactive MS lesions [104], indicating that astrocytes may act as target of fingolimod and siponimod within the CNS. Induced S1Pr expression levels have as well been reported in the experimental autoimmune encephalomyelitis model [105, 106]. Interestingly, it has been suggested that overexpression of the S1Pr1 on reactive astrocytes drives neuropathology of MS rebound after fingolimod discontinuation [107]”.
Q3: Does the tolerance develop after prolonged siponimod treatment?
A3: This is indeed an interesting point. I am not aware of any study which observed tolerance after prolonged siponimod exposure.
Q4: Is S1Pr1 the dominant receptor attributing to the therapeutic effects of siponimod? More supporting evidence should be provided.
A4: Most studies suggest that the S1Pr1is the dominant receptor attributing to the therapeutic effects of fingolimod and, probably siponimod. However, own not yet published data clearly show that the protective effect of siponimod in mediated via the S1Pr5. In this study we investigated protective functions of siponimod in cuprizone-intoxicated mice. This model is characterized by metabolic oligodendrocyte injury without major roles of lymphocytes. During the cuprizone intoxication period, mice were daily treated with either vehicle or siponimod (3.125mg/kg) via oral gavage. RAG-/- and S1PR5-/- mice were used to study the relevance of T/B-cells and the S1PR5-receptor. We were able to show that while RAG-deficient mice are still protected by siponimod, this protective effect is absence in S1PR5-/- mice. We hope that we will publish these interesting findings in the near future.
Q5: What is the advantage of using siponimod compared with FTY720?
A5: Thank you for this comment. This is indeed an interesting and important point, and we obviously were not able to make this aspect clear in the first draft of the manuscript. Thus, we now state in the final conclusion “Potential advantages of using siponimod compared with fingolimod are its greater receptor specificity, more stable and predictable kinetics (due to the fact that siponimod is not a pro-drug), and, most importantly, an approved beneficial effects of siponimod in a SPMS clinical study.”
Q6: What are the common adverse effects of siponimod in MS patients?
A6: Thank you for this common. Although this is a highly relevant question we feel that this is out of the scope of this review article. We hope the kind reviewer can follow our argumentation.
Q7:Please use tables to summarize the findings of siponimod action in various cell types, as well as its therapeutic effects in the MS animal models and human patients.
A7: There are several reports which have addressed the mode of action and the cellular targets of fingolimod, and plenty of reviews (too many I should say) have addressed this point. By far less is known about the cellular targets as well as the mode of action of siponimod. We have tried to summarize these effects in the text and hope it will be valuable for the interested reader in the present form. For a table, we think, it is too early.
Q8: Which brain area is shown in Figure 1? Is there any reference (s) for the images presented?
A8: This image is derived from MS samples obtained in collaboration with the Netherlands Brain bank, not yet published elsewhere. It shows a part of the centrum semiovale of an MS patients. We now state in the figure legend “Paraffin-embedded postmortem brain tissues were obtained through a rapid autopsy protocol from donors with mainly progressive MS in collaboration with the Netherlands Brain Brank, Amsterdam. The study was approved by the institutional ethics review board, and all donors or their relatives provided written consent for the use of brain tissues and clinical information for research purposes.”
Reviewer 2 Report
This review examines the question “Does Siponimod exert direct effects in the central nervous system?” and concludes that there is sufficient evidence to show direct CNS effects. Furthermore the author proposes that it may be due to these direct effects that Siponimod is effective for treatment of multiple sclerosis, unlike FTY720/Fingolimod. The author presents a clear, balanced and scientifically accurate description of the field, accounting for the diverse and complex effects of Sphinogosine-1-phosphate signalling and the intereference thereof on peripheral immune functions. It is clear from more recent studies that S1P signalling is also important within the CNS and direct interference with this may be efficacious within the context of MS. In summary this article represents a timely and novel review of the current literature on this subject and would appeal to both specific and more general audiences interested in MS, neuroinflammation or similar disciplines. I would recommend accepting this article for publication in its current form.
On a small copy-editing note line 249 spleen is misspelt splen
Author Response
We would like to thank the kind reviewer for the fine evaluation of our work.